# Environmental Impacts of University Restaurant Menus: A Case Study in Brazil

**Maria Hatjiathanassiadou [1], Sthephany Rayanne Gomes de Souza [1], Josimara Pereira Nogueira [2], Luciana de Medeiros Oliveira [3], Virgílio José Strasburg [4], Priscilla Moura Rolim [1]** 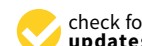 **and Larissa Mont'Alverne Jucá Seabra [1,***

[1] Department of Nutrition, Federal University of Rio Grande do Norte, Av. Senador Salgado Filho, s/n, Lagoa Nova, Natal 59078-970, RN, Brazil; mariahatji@hotmail.com (M.H.); theiagomess@gmail.com (S.R.G.d.S.); priscillanutri@hotmail.com (P.M.R.)

[2] Nutrition Post Graduate Program, Federal University of Rio Grande do Norte, Av. Senador Salgado Filho, s/n, Lagoa Nova, Natal 59078-970, RN, Brazil; josinogueiira147@hotmail.com

[3] Federal University of Rio Grande do Norte, University Restaurant, Av. Senador Salgado Filho, s/n, Lagoa Nova, Natal 59078-970, RN, Brazil; luciana_medeiros_@hotmail.com

[4] Department of Nutrition, Universidade Federal do Rio Grande do Sul, Rua Ramiro Barcelos, n° 2400, Santa Cecília, Porto Alegre 90040-060, RS, Brazil; virgilio_nut@ufrgs.br

* Correspondence: larissaseabra@yahoo.com.br

**Abstract:** The production of collective meals in institutional restaurants demands a great use of natural resources. The search for strategies to reduce negative environmental impacts in this sector is essential to offer meals that are not only healthy but also sustainable. In this study the evaluation of water footprint (WF) of menus offered in a public university restaurant located in the northeast of Brazil and the verification of the origin of foodstuff purchased to compose the menus in 2 months were carried out. The study is transversal, descriptive, and exploratory and the data were collected between March and April 2018. Water footprint of 112 traditional and vegetarian menus was calculated and the results showed that the WF of traditional menus was higher ($p < 0.0001$) than the vegetarian menus. Weekly average per capita of WF was 2752.0 L for traditional menus and 1113.9 L for the vegetarian option, with animal protein intake in the traditional version being the main factor for this difference. It was identified that 49.47% of the foods used in the studied period originated from the same state where the restaurant is located, with fresh vegetables being the food group with the highest contribution. Thus, we conclude that although the restaurant purchases local food products, the environmental impact of the choice of foods that is included in the menus must be taken into account. The utilization of local foodstuff and the decrease of animal protein in the menus can be a good strategy to encourage sustainable actions in food services meal production.

**Keywords:** foodservices; environmental impacts; water footprint; sustainability

## 1. Introduction

Despite the fact that over the past 50 years, intensification of agriculture has been responsible for the rapid growth of the food supply, interfering positively in reducing hunger and improving the overall nutrition picture, this has brought high costs to the society [1]. The consequences can be seen in rivers, lakes, and oceans in the form of chemical pollution; in the soil by its degradation and loss of fertility; in the health of the population and the development of the environment; and on the climate change and global warming by the production of high levels of greenhouse gases [1]. Such knowledge suggests changes in food habits as an alternative to reducing the environmental degradation caused by the production system [2].

Considering the expansion scenario of goods production and consumption patterns, sustainability arises as a necessary theme, mainly due to the perspective of the finiteness of natural resources caused by the naturalization of the exaggerated practice of these activities [3]. In this perspective, sustainable development is being established worldwide through the use of strategies, plans, and policies, dissolved in the economic, social, and environmental aspects of the society, to allow the resources used to suffice for present and future generations [4–6].

Within the western diets, the difference between the amount of energy consumed in the form of food and the amount of energy and resources needed to produce it is large [7]. Although the intensification of this area in the recent years has reduced hunger and improved the overall nutrition framework, this progress has brought high costs to the society [1].

The 2030 Agenda for Sustainable Development approach developed by the United Nations Organization (2015), encompasses 17 goals with action plans based on people, the planet, prosperity, peace, and partnership. Some of the goals presented are applicable in the production of meals and collective feeding, as they seek to end hunger and promote sustainable agriculture (objective 2) and to ensure sustainable production and consumption patterns (objective 12) [8].

Only in Brazil, the collective food market earned about R$50.8 billion and served approximately 20.5 billion meals in 2018, demonstrating the great economic, environmental, and social impact that this sector currently has [9]. The institutional dining spaces in private or public companies are among the various types of catering services offered to the community.

University restaurants (UR) of public educational institutions have the main objective to feed students and staff, indirectly helping to reach the final objective of the institutions, which concerns the training of professionals in all areas. They are also presented as a study environment for the development of practices that enhance their service and generate not only economic but also social and environmental impacts [10,11].

In this perspective, sustainable practices in meal production are of great importance, given the greatness of the market, and consequently, its potential impact on the environment in a positive or negative way. In the production of collective meals, there are many processes that cause economic and environmental impacts and the use of indicators is one of the ways to evaluate sustainable practices in foodservices [12]. One of the indicators that can be used in food production and also in the evaluation of menus in foodservice is the water footprint (WF). WF estimates the total volume of direct and indirect use of fresh water to produce the goods and services consumed by an individual or community [13].

However, sustainable practices in meal production in food services are based on the environmental issues related to waste generation, disposal of products and packaging, and the use of large amounts of water during the various stages of the production of meals, for example [10]. It should be considered that the term sustainability also encompasses economic and social areas, through the strengthening of the local economy, the acquisition of family farming foodstuff, and the right to adequate and safe food [14]. Thus, this study aims to evaluate the environmental impacts through: (a) Water footprint estimation of conventional and vegetarian menus; and (b) identification of the place of origin of foodstuff used in meals served in a university restaurant of a federal public university in Brazil.

## 2. Literature Review

### 2.1. Collective Meal Production

The provision of meals outside home in communities occurs in specific spaces, and the term used to characterize this activity is defined as foodservice. In Brazil, institutional foodservices are known as food and nutrition units (FNU) and are identified as units that manage both technical, administrative, and distribution activities for healthy and sick groups, with the main objective of "contributing to maintain, improve, or recover the health of the clientele served" [15]. In general, FNU are usually classified into two types: (1) commercial—which include restaurants, cafeterias, cafes, pubs, among

others; and (2) institutional—covering care for schools, hospitals, companies, military institutions, among others [16].

Within a FNU, the actions must be dynamic and interactive, based on the administrative process, which include: Planning the objectives to be achieved; decision-making for future action and determination of plans; the organization, defining, and dividing the attributions, areas, and works to be exercised; the direction, associated to the command and supervision of human and material resources; and the control, constituted by the establishment of standards or criteria, performance observation as well as its comparison with the established standard and the application of corrective actions to correct errors and variations [17].

In this context, professionals who work in segment of food for the community should be aware of the scope of the internal functioning of their place of operation. There is an interface that also involves a broader look and action that should consider the sustainability aspects such as the rational use of natural resources and the origin of the raw materials that will be used to provide the meals [10,17].

## 2.2. Foodstuff Origin

Sustainability in the area of collective nutrition is based on the environmental issues related to the generation of waste, the disposal of products and packaging, and the use of large amounts of water during the various stages of food production, for example [12]. It should be stressed that the sustainability triple bottom line also incorporates the economic and social areas, which include strengthening the local economy, acquiring family farming supplies, and the right to adequate and safe food [14].

In order to encourage the development of responsible practices that conserve natural resources, the American Dietetic Association (ADA) presents strategic guidelines that can be used by the professionals involved in food production, in order to consider all the food system (production process, transformation, distribution, access, and consumption). The document entitled "Position of the American Dietetic Association: Food and Nutrition Professionals Can Implement Practices to Conserve Natural Resources and Support Ecological Sustainability" recommends the adoption of practices by the professionals responsible for collective meal production with the aim of encouraging a sustainable food and nutritional system. Among the recommendations we can mention the offer of a variety of food choices, increasing the purchase of food produced locally and reducing the purchase of imported food, in addition to minimizing food waste [12].

Community food systems can be described as collaborative efforts to build more locally based, self-reliant food economies in which sustainable food production, processing, distribution, and consumption are integrated to enhance the economic, environmental, and social health of a particular place. This position is particularly concerned with how community food system building can serve as a strategy to improve or maintain the environmental health of localities [18].

The Food Guide for the Brazilian Population, in support of Food Security, raises criticism regarding food choices when it includes in its discourse adequate and sustainable productive practices. The food/nutrient supply to the population was not the only highlight, the authors affirm that the recommendation of these should also take into account "the impact of food production and distribution on justice and environmental integrity," once the concentration of rural properties in the hands of large entrepreneurs is observed, as well as the interference of farmers' autonomy in choosing the seeds, fertilizers, and forms of pest and disease control; number of intermediaries between farmers and consumers; the techniques applied to soil conservation; the planting of transgenic seeds; the form of pest and disease control; means of transport and intensive and extensive forms of animal husbandry [19].

The consumer purchasing food at an institution should take into account not only the price or profit of the establishment, but also the strengthening of agriculture and the local producer. The choice of food and its processing are the issues to be addressed with the objective of not only taking care of the health, but also taking responsibility for the social and environmental issues involved [19,20]. The

insertion of family agriculture is relevant, since it is embodied as a strategy both in relation to food and nutritional security and the promotion of sustainability in the areas of food and agriculture, allowing the promotion of biodiversity; detention and reversal of land degradation, as well as combating desertification; poverty reduction and rural inequality [1,20]; issues that are present in the United Nations proposed Sustainable Development Objectives [8].

According Pérez-Mesa et al. [21], the efficient management of agri-food supply chains is a key concept for the agri-food sector's competitiveness, while also generating increasing interest among researchers and practitioners because of the growing demand for high quality, fresh (local), value added, and customized agri-food products. The authors highlighted in their work various supply chain management strategies of retail distribution companies, focusing on the most prominent firms in Europe and on perishable products. The results showed that social and environmental management of production have been relegated to secondary status, despite being key aspects for establishing sustainable supply chains, particularly considering that consumers want to know that their purchases generate positive environments and social externalities.

### 2.3. Water Footprint

The concept of water footprint is defined as: "an indicator of water use that considers the direct and indirect use of water from a consumer or producer." The water footprint of an individual, community, or company is defined as the total volume of fresh water used to produce the goods and services consumed by the individual or community or produced by the company. Water use is measured in terms of volumes of water consumed (evaporated) and/or polluted per unit of time [13].

Within this concept of water footprint, there is also the term known as "virtual water." The concept was first introduced by John Anthony Allan in 1998, having been defined as water incorporated into commodities. It refers to the consumption of water by humans not only associated with direct consumption, but is also incorporated in products consumed, in addition to the portion used during production, manufacturing, and transportation, i.e., water involved in the production process [22–24]. The water footprint is used as an indicator to map the impact of human consumption on global freshwater resources, and is termed as a geographically explicit indicator, as it does not only present volumes of water used and pollution caused, but also the locations [13,24].

This indicator is expressed in volume per unit of product (usually expressed in L or $m^3$/ton), the sum of the water footprints of the process steps being taken to produce the product [25]. According to Bleninger and Kotsuka [24] and Mekonnen and Hoekstra [25], Hoekstra et al. [26], some limitations of using this indicator are the difficulty to find all the necessary data for the calculations of the water footprint and also the fact that the tool only focuses on the analysis of fresh water, and does not include other aspects such as climatic changes or social and/or economic aspects.

Water is a precious natural resource and should be used sustainably [27]. In addition to being essential for human survival from a physiological point of view, it is also an essential component for economic activities such as agricultural and industrial operations [28]. However, production systems and the current consumer market are often environmentally, socially, and economically unsustainable. In this scenario, drinking water consumption is one of the major environmental impacts caused by activities such as agriculture and food production [29]. Thus, it is important to use indicators to support sustainable practices in the production of meals and encourage healthy and sustainable food practices.

In a study developed by Strasburg and Jahno [30], the sustainability of menus was evaluated in a university restaurant in the city of Porto Alegre, Brazil. The evaluation was made by quantifying the water footprint in meals served, including lunch and dinner. An WF average of 2 L was recorded for each prepared meal, with the highest contribution from beef. For the authors, the evaluation of WF when applied in meals can be a possibility for the rational use of natural resources of the planet and for the development of eating habits and consumption. The same authors also proposed the calculation of eco-efficiency in the use of raw materials for the production of meals in which the WF was one of the evaluated items [31].

Plant-based diet provides a significant water conservation benefit and many societies, governments in particular, will have to reconsider the increasing demand for an animal-based diet [32]. ADA document describes that meat protein requires more land for production than vegetable protein. In addition, meat protein production required approximately 26 times more water than vegetable protein on rainfed lands, and production of vegetable proteins was 2.5 to 50 times more energy efficient than meat production [12].

Willet et al. highlighted that methodological inconsistencies and data gaps make it difficult to distinguish and compare the precise environmental footprints of individual food products. However, results from a large and growing body of literature points toward a very likely clear hierarchy of impacts among larger food categories. According to the authors a universal healthy reference diet consists of vegetables, fruits, whole grains, legumes, nuts, and unsaturated oils, includes a low to moderate amount of seafood and poultry, and includes no or a low quantity of red meat, processed meat, added sugar, refined grains, and starchy vegetables [33].

## 3. Methodology

### *3.1. Study Characterization*

This case study is a cross-sectional, descriptive, and exploratory study carried out at the University Restaurant of the Federal University of Rio Grande do Norte (UFRN), Brazil. Data collection was carried out in the months of March and April of 2018.

The University Restaurant of UFRN is responsible for the supply of about 4000 meals/day, for students and university employees and is located in the Northeast region of Brazil (Figure 1). It is important to highlight that the Northeast, despite breaking a tradition and showing a good period of growth with simultaneous social improvements, remains less developed than others regions of the country [34]. Thus, public university restaurants play an important social role.

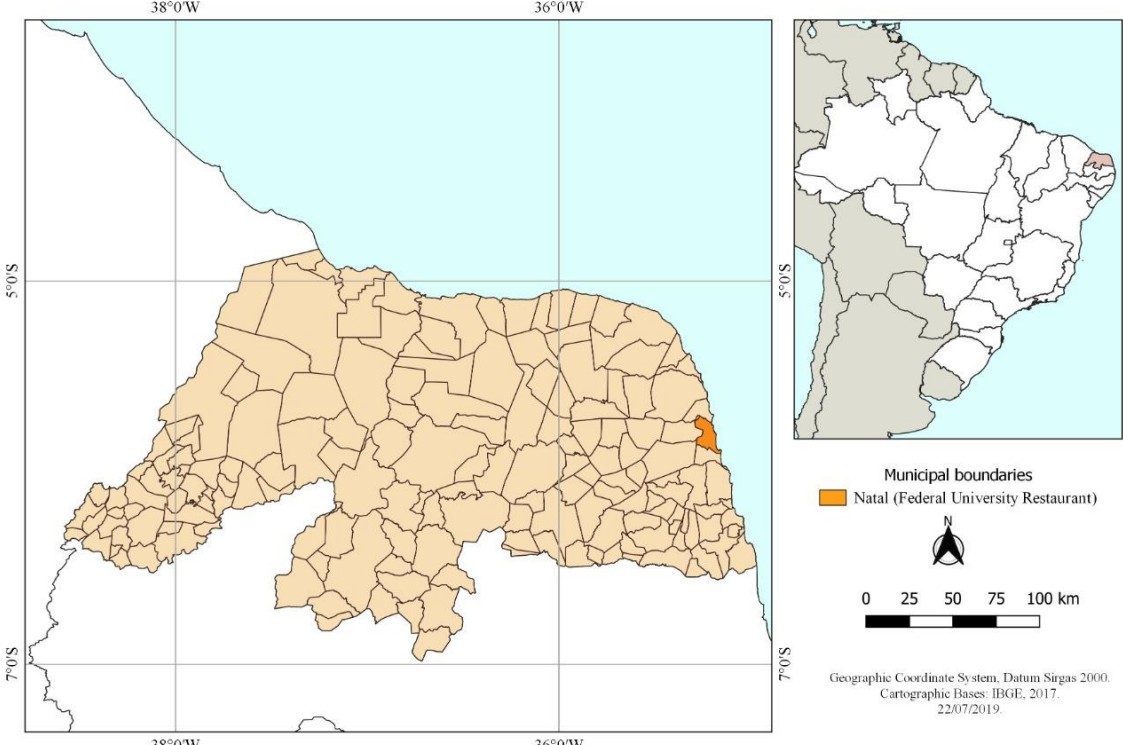

**Figure 1.** Map of study area: location of Federal University of Rio Grande do Norte, Natal, Brazil.

Breakfast, lunch, and dinner are provided every day of the week. The central unit is responsible for producing both meals that supply its unit, as well as providing weekly and monthly kits to resident students, for basic supplies. The lunch menu consists of entrees (salads, soups and other regional hot dishes), main dishes (beef, pork, poultry, and fish), base dishes (rice and beans), accompaniments (*farofa* crumbs, tubers, couscous, etc.), juices based on fruit pulp, and dessert (sweet pastries or sugary fruit). The restaurant also has vegetarian preparations, made mainly from milk and dairy products, cereals, legumes, and vegetables.

The university restaurant has the per capita control of all meals offered. This control is done through standardized technical cards organized in a "digital book," divided by preparations according to the constitution of the menu. This control is important because it assists both in the process of acquiring foodstuff and in the process of preparation of meals, to avoid the lack or excess of food, which can result in waste. The stock of foodstuff destined to the preparation of the meals is administered by the central warehouse, with control of entrance and exit of perishable items realized by nutritionists.

### 3.2. Data Collection

### 3.2.1. Origin of Foodstuff

For data collection regarding origin of foodstuff, two months purchase invoices were analyzed. Additionally, nutrition labeling information of foodstuff was collected by a cellular device camera for further evaluation. For foodstuff that were produced in more than one country, the country of origin considered was the country in which the product received its last substantial processing.

Foods were classified into 12 groups: (1) vegetables, greens, herbs and spices; (2) cereals; (3) fruits; (4) legumes; (5) sugars; (6) vegetable oils and fats; (7) infusions and beverages; (8) sauces; (9) other ingredients (ingredients that added color, flavor and texture to meals like vinegar, meat tenderizer and all kinds of salts); (10) milk and dairy; (11) eggs; (12) meat products.

Some in nature foods did not have labeled packages (vegetables, fruits, and some meats) and their origin did not appear in the invoice. In this case, the place of food origin was verified by analysis of the food production in the state of Rio Grande do Norte through official documents [35,36] and observation of the supplier's information in the invoices. The foodstuff was considered as "State" when produced in Rio Grande do Norte, state where the meals were produced and distributed; "Regional" foods were produced in the Northeast region of Brazil; "National" foods were produced in the other regions of Brazil, not in the Northeast; and "International" foods were produced in other countries.

### 3.2.2. Water Footprint of Menus

For the analysis of WF, meals cooked on each day of the month were considered individually and each ingredient and/or food used was evaluated in all preparations served in the lunch menu. The menus of the traditional and vegetarian lines were analyzed separately. For this purpose, information on per capita in grams of each ingredient/food, number of meals served, total amount served in grams and total estimated WF, and estimated WF per capita were used. The values proposed by Hoekstra [13], Hoekstra [26], Mekonnen and Hoekstra [25], Pahlow et al. [37] were used as reference to estimate meals WF (Appendix A). For foods that do not have water footprint values available in the literature, it was decided to use food values from the same group or ingredients.

### 3.2.3. Water Footprint of Food Groups

The percentage of contribution of the water footprint by food groups was evaluated, also related to the food supply in kilograms (kg). Four food groups were instituted: (1) animal origin; (2) vegetable origin (including legumes, vegetables, cereals, and fruits); (3) oils and fats; (4) sugars and sweets. These groups were chosen from the analysis of the menu and evaluation of the most frequent products. For this evaluation, the values of the amount used in kg in the food menu in each of the groups and the value of the water footprint of the foods present in each group were considered.

### 3.3. Data Analysis

The data were analyzed through the program Microsoft Excel® version 2013, with results demonstrated by calculating the absolute frequency and percentage for the evaluation of the origin of foodstuff. For analysis of WF of the menus, Shapiro–Wilk test was carried out to verify data normality and after that T-test was performed to compare the average water footprint of both menus. Significance level adopted for all tests was 0.05 ($p < 0.05$) and the analyzes were performed using Action 3.0 software (Estatcamp, Sao Carlos, Brazil).

## 4. Results and Discussion

### 4.1. Evaluation of Foodstuff Origin

Table 1 presents the origin of the foodstuff used for the lunch meal purchased in the University Restaurant according to its category. In all of the items purchased, the most frequent was the purchase of state products, followed by products from other parts of Brazil. It was observed that the category of vegetables and fruits, milk, and dairy products obtained the largest number of state products. The regional ones prevailed in the category of infusions and drinks. Nationals were heavily represented in the meat, sauces, and cereals groups. Only one item of international origin was identified in the legume category (1.06%).

**Table 1.** Origin of foodstuff, by category, purchased by the Federal University of Rio Grande do Norte (UFRN) restaurant in March and April 2018.

| Category | State | Regional | National | International |
|---|---|---|---|---|
| Vegetables | 29 | 3 | 6 | 0 |
| Cereals | 1 | 1 | 7 | 0 |
| Fruits | 6 | 0 | 2 | 0 |
| Legumes | 3 | 1 | 1 | 1 |
| Sugars | 1 | 0 | 0 | 0 |
| Oils and vegetable oils | 0 | 0 | 1 | 0 |
| Infusions and beverages | 0 | 6 | 0 | 0 |
| Sauces | 2 | 1 | 5 | 0 |
| Other ingredients | 2 | 1 | 2 | 0 |
| Milk and dairy | 1 | 0 | 2 | 0 |
| Eggs | 1 | 0 | 0 | 0 |
| Meat products | 1 | 2 | 6 | 0 |
| Total % | 49.47 | 15.79 | 33.68 | 1.06 |

Regarding to products of animal origin, which had a total acquisition of more than 42 tons, it can be seen that eggs and part of the beef were of regional origin. It is also observed that chicken cuts, beef, and pork products were from the northeastern region of Brazil and other Brazilian states (Figure 2). Products of processed vegetable origin present a particular situation because they comprise a variety of products such as sugar, oil, sauces, fruit pulp, and others that have different origins. However, it is worth mentioning the use of salt that individually had the largest acquisition and the state of RN is the main producer of the product in Brazil.

Fresh vegetables comprise a greater variety of items such as vegetables and leafy ones, fruits, cereals, and beans. These products are used as the main energy suppliers of the menus in the university restaurant. They added more than 48 tons of consumption during the investigation period. Of this, 80.64% of the quantity acquired were from Rio Grande do Norte State.

The purchase of products of national origin, especially meat products, sauces, and cereals can be explained by the centralization of production and distribution, now dominated by large companies. This reality is built in the face of population growth and urbanization, where increasing production and supply in sufficient quantities are necessary. However, the use of techniques that are adapted to weaken the quality of the food and the natural and social environment are not taken into consideration.

The defense of production and local practices are based on the premise that the valorization of small production and the use of traditional techniques can promote sustainability. So, it is necessary to set up initiatives through central and local governments and producer associations, and it is important to be involved with the community itself, environmental associations and the public sectors [38].

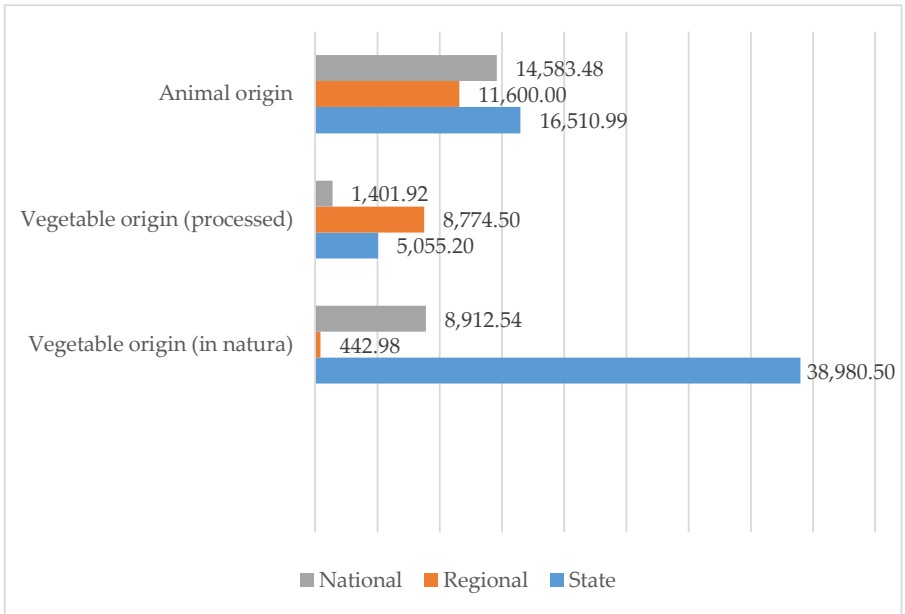

**Figure 2.** Foodstuff by category, purchased by the UFRN Restaurant, Natal (RN), in March and April 2018.

Brazilian public institutions that purchase foodstuff are encouraged to buy at least 30% of the food from family producers. Through the Institutional Purchase Food Purchase Program, public agencies can purchase food from family farms through public calls with their own financial resources [20]. It is a public policy that uses the State's purchasing power to promote local growth and income and also guarantees the population the right to adequate and sustainable food.

The diverse systems involved in global food production account for a significant share of total greenhouse gas (GHG) emissions. These emissions are generated in the stages of production and transportation of the raw materials to the procedures used during the processing of the final products [39,40]. Weber and Matthews [41] in a study of climate change and food choices in the United States found that there will be a 4 to 5 percent reduction in greenhouse gas emissions if food purchases are from local producers. This is because in food production there are large emissions of carbon dioxide ($CO_2$) and other gases. The study also estimated that a family diet based entirely on local produce will reduce GHG emissions equivalent to 1600 km/year. The reduction is more significant when the consumption of red meat and dairy products for chicken, fish, and eggs or a vegetable-based diet (8590 km/year and 13,000 km/year, respectively) are substituted.

Michalský e Hooda [42] adds that local production is also an alternative to the substitution of imported and transported fruits and vegetables by air, considering that it requires fewer energy inputs, also reducing the excessive emission of $CO_2$ and other harmful gases to the environment. The authors believe that the transition to local fruit and vegetable production in the United Kingdom will be conducive to meeting Agenda 2030, bringing positive impacts not only to their country but also globally. Smith et al. [43] pointed out in their study that in Berlin (Germany) the use of locally produced food is not encouraged by the city's procurement policy. De Laurentiis et al. [44] in a study on the amount of GHG emissions related to input transport have identified that inputs purchased within the United Kingdom have shown reduced emission values when purchased during the harvesting period compared to products of external origin.

### 4.2. Evaluation of the Water Footprint

Table 2 presents weekly per capita WF of the two types of menus served daily in the university restaurant of UFRN. Comparing the results of the traditional and vegetarian menus, it can be seen that the per capita value of WF was significantly higher ($p < 0.0001$) in the traditional menu. This is due to the variations of the items in the menu, mainly concerning the main course (meats); the weeks in which more meat-based preparations were served presented higher values compared with the weeks with a greater offer of pork and chicken. It was evidenced in the present study with a water footprint 2.47 times higher in the traditional menu, which is directly associated with the use of animal products, mainly beef.

**Table 2.** Per capita average water footprint (L/Kg) of traditional and vegetarian lunch menus of a Brazilian public university restaurant.

| Week | WF Traditional Menu | WF Vegetarian Menu |
|---|---|---|
| 1 | 2517.2 | 992.3 |
| 2 | 1991.9 | 1121.5 |
| 3 | 3024.2 | 970.2 |
| 4 | 2465.1 | 1224.8 |
| 5 | 3144.7 | 1341.4 |
| 6 | 2859.0 | 1156.1 |
| 7 | 3099.2 | 1021.3 |
| 8 | 2915.0 | 1083.9 |
| **Average \*** | 2752.4 [a] (SD 396.8) | 1113.9 [b] (SD 125.8) |

\* average with different letters show statistically significant difference ($p < 0.05$) by T-test.

Comparing the food groups based on amount in kilograms and the WF, it was observed that foods of animal origin have a higher WF value considering the proportion of the supply in kilograms of these foods. The inverse situation between the amount in kg and WF of the food of plant origin is shown in Figure 3.

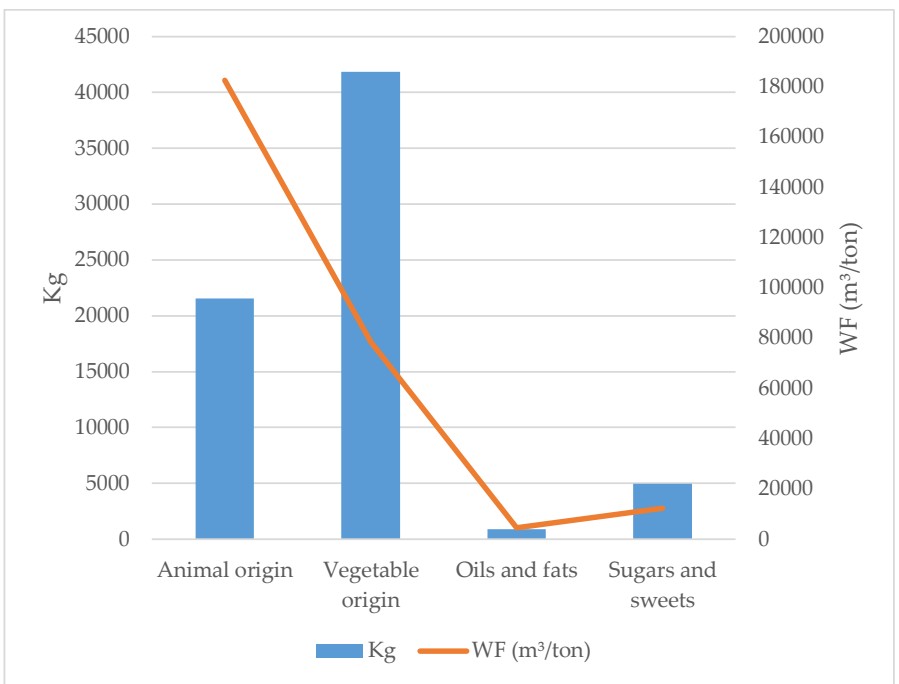

**Figure 3.** Relationship between food weight and water footprint by food groups of the university restaurant menu. Natal (RN), in March and April of 2018.

It is increasingly common to find scientific studies that seek to investigate the environmental impact of the water footprint of traditional diets that include meats and animal foods compared to other types of diets, such as vegetarian and vegan. This study further reinforces that meat has a major environmental impact, especially when compared to other food sources. Mekonnen and Hoekstra [45], showed that from a freshwater resource perspective, it is more efficient to obtain calories, protein and fat through crop products than animal products. Animal products generally have a higher water footprint than crop products. The average water footprint per calorie for beef is 20 times larger than for cereals and starchy roots.

According to a study by De Laurentiis, Hunt, and Rogers [46], carried out in the schools in the United Kingdom to evaluate the environmental impact of the menus, it was noticed that when comparing the weight of the meal with the environmental impact, the meat items contributed the highest WF value, especially those of bovine and ovine origin, compared to poultry and pork meats. Regarding the menus in the UK and UFRN, this verification was also confirmed by comparing the WF values of the traditional and vegetarian menus.

De Laurentiis, Hunt, and Rogers [46] also evaluated the positive impact of vegetarian and vegan diets, as they bring less impact to the environment, and are considered more positive for individuals' health since they have development protection factors of chronic noncommunicable diseases (CNCDs). Despite this, there is still much debate about how healthy and nutritionally adequate the vegetarian diet is, especially about concerning adequate protein intake. According to the American Dietetic Association [47], vegetarian diets are healthy, nutritionally adequate, and promote various health benefits in the prevention and treatment of certain diseases.

A study conducted by Strasburg and Jahno [30] evaluated the WF of menus of a university restaurant in Porto Alegre/RS (Brazil). The animal food group accounted for 77.9% of the WF, with beef and chicken being responsible for 62.2% of this value. On the other hand, those of vegetable origin provided the highest amount in kilograms of food (65.5% of the total) with a WF of 21.2%. These results are similar to those found in the present study.

The daily average per capita WF of European diets is 4.265 L/day. The value found in the present study was 2752.4 L/day per capita only in the traditional menu lunch, thus evidencing the high WF value for the production of meals of this type of menu. Moreover, the vegetarian lunch presented WF of 1113.9 L/day. The high WF value observed especially in the traditional menu can be explained by the fact that the portion of animal protein served in the restaurant is high (around 200 g per person). It is also possible to observe that 53% of the contribution of WF per capita of European food comes from products of animal origin (2290 L/day), with beef, pork, and milk contributing the most, reaffirming the great environmental impact of these products [48].

It is possible to perceive the positivity of the greater inclusion of food of vegetal origin in the menus, since these have better environmental performance when compared to the ones of animal origin, especially the meats. With respect to the Brazilian per capita, an average of 1.992 L/day of food outside the home based on the Family Budget Survey 2008–2009 is used [49]. Still, in this study, of the 26 evaluated items (6 of animal origin and 20 of vegetal origin), those of animal origin accounted for 63.3% of the value of water consumption, accounting for only 31.2% of consumption in grams. Strasburg and Jahno [31] found a relationship between the value of animal products used, especially beef, with environmental performance in Brazilian's restaurants.

Hölker et al. [50] emphasizes that it is necessary to significantly reduce the consumption of animal source foods for reasons like animal welfare, human health, and environmental issues. The production of animal source foods is, across various environmental indicators, much more harmful to the environment than the production of plant-based food [51]. According Willett et al. [33], a significant reduction (i.e., 50–75%) in the consumption of animal source foods is usually proposed, especially for developed countries.

Graham et al. [52] found that sandwiches containing beef and cheese had the greatest environmental impact; in contrast, the vegan dishes required the least amount of water and land. Furthermore, the

authors demonstrated a clear gradient in sandwich environmental impact according to meat type filling, with beef having the greatest impact followed by pork and chicken. However, analysis of the nutrient content of these options has revealed that choosing options with a lower environmental impact may have health benefits in terms of reducing calorie and sodium intake but have fewer micronutrients such as iron. Thus, managers of institutional foodservices have the challenge of finding a balance between producing sustainable and nutritionally adequate meals.

Studies that seek to evaluate other forms of impact, such as the carbon footprint, also prove the great impact that diets containing meat have on the environment when compared to alternative diets. Cerutti et al. [53] in a study on the school feeding service of the city of Turin compared the emission of GEE between the traditional menu with the vegetarian menus and another without the red meat. The menu without red meat had a 32% reduction and the vegetarian menu had a 54% reduction in GHG emissions.

## 5. Final Considerations

The present study showed that between two types of menus offered in a public university restaurant, the conventional standard presented the higher water footprint because of the use of products of animal origin in its composition. Regarding the origin of foodstuff, most of them originated in the state where the restaurant is located, which can be considered as a positive factor in the search for sustainable food production.

The high meat consumption culture in institutional restaurants needs to be reviewed not only when thinking about health but also the environment. The use of indicators such as water footprints in institutional restaurants can serve as a basis for educational actions and public policies that prioritize the supply of foods that have a lower negative environmental impact. It should also be stressed that it is important to carry out the evaluation and constant monitoring of the environmental impacts of the use of inputs and the activities related to the elaboration of menus and the provision of collective meals.

In this research the following aspects can be considered as limitations: the investigation was limited to one restaurant and menus evaluated in a period of two months. Identifying foodstuff origin demanded the evaluation of food labels and invoices of food acquired during the period and these activities required exhaustive work, which makes it difficult to evaluate it for a longer period. However, it will be interesting to carry out the evaluation of menus for a greater number of institutional restaurants, since sustainable practices, such as cost management and compliance with food safety legal requirements, are generally neglected in these places. The study can be replicated in the context of other educational institutions and also in the scope of companies that work in the segment of collective meals.

The results of the research offer subsidies so that the managers of foodservices can better plan actions related to the organization of the menus offered, aiming at reducing the environmental impact of meal production. However, without government involvement in strengthening public policies that support the adoption of sustainable practices by the food sector, the scope of this goal becomes more distant.

**Author Contributions:** Conceptualization: L.M.A.J.S., V.J.S., and P.M.R.; data collection: L.M.A.J.S., M.H., S.R.G.d.S., L.d.M.O., and J.P.N.; writing—original draft: L.M.A.J.S., P.M.R., M.H., and S.R.G.d.S.; writing—review and editing: L.M.A.J.S., M.H., J.P.N., and V.J.S.

**Funding:** This research received no external funding.

**Acknowledgments:** The authors would like to thank the University Restaurant of Federal University of Rio Grande do Norte and the UFRN Pro-Rector for Research (PROPESQ) for their support through the Scientific Initiation Program; and Aurino Alves Nunes Filho for the construction of the map of Rio Grande do Norte state.

**Conflicts of Interest:** The authors declare no conflict of interest.

# Appendix A

**Table A1.** Water footprint values in kg/L for each food item used.

| Foodstuff | Unit | GAF * | Foodstuff | Unit | GAF * |
|---|---|---|---|---|---|
| Bacon [4] | 1 kg | 4800 | Watermelon [1] | 1 kg | 235 |
| Banana [2] | 1 kg | 860 | Melon [4] | 1 kg | 235 |
| Beans [1] | 1 kg | 5053 | Butter [4] | 1 kg | 5000 |
| Beef [2] | 1 kg | 15500 | Worcestershire sauce [4] | 1 kg | 613 |
| Cabbages and other brassicas [1] | 1 kg | 280 | Frozen guava pulp [4] | 1 kg | 1800 |
| Carrot [1] | 1 kg | 195 | Peach or nectarine [2] | 1 kg | 1200 |
| Cashew nuts [1] | 1 kg | 14218 | Wheat bread [2] | 1 kg | 1300 |
| Cassava flour [1] | 1 kg | 1872 | Shoyu [4] | 1 kg | 613 |
| Cauliflowers and broccoli [1] | 1 kg | 285 | Frozen mango pulp [4] | 1 kg | 1800 |
| Chayote [4] | 1 kg | 353 | Frozen acerola cherry pulp [4] | 1 kg | 413 |
| Chicken [2] | 1 kg | 3900 | Grape [1] | 1 kg | 608 |
| Chickpeas [1] | 1 kg | 4177 | Wine [2] | 1 glass of 125 mL | 120 |
| Chive [4] | 1 kg | 8280 | Cucumber or pumpkin [2] | 1 kg | 240 |
| Chocolate milk [4] | 1 kg | 15363 | Milk [2] | 1 glass of 250 mL | 250 |
| Chocolate [1] | 1 kg | 24000 | Bell pepper [1] | 1 kg | 379 |
| Cinnamon [1] | 1 kg | 15526 | Soybean oil [2] | 1 kg | 4190 |
| Cocoa powder [1] | 1 kg | 15636 | Mustard [4] | 1 kg | 2809 |
| Coffee [2] | 1 cup of 250 mL | 140 | Sunflower seed oil [1] | 1 kg | 6792 |
| Coriander [1] | 1 kg | 8280 | Frozen cashier pulp [4] | 1 kg | 3793 |
| Dried peas [1] | 1 kg | 1979 | Cabbage [2] | 1 kg | 200 |
| Garlic [1] | 1 kg | 589 | Bay [4] | 1 kg | 8280 |
| Green Beans [1] | 1 kg | 561 | Wheat for kibbe [4] | 1 kg | 2035 |
| Guava jam [4] | 1 kg | 1800 | Okra [1] | 1 kg | 576 |
| Lettuce [2] | 1 kg | 130 | Lemon [1] | 1 kg | 642 |
| Maize (corn) starch [1] | 1 kg | 1671 | Mayonnaise [4] | 1 kg | 4190 |
| Maize flour [1] | 1 kg | 1.253 | Tapioca paste [4] | 1 kg | 2818 |
| Milk cream [4] | 1 kg | 4600 | Texturized soy protein [4] | 1 kg | 2145 |
| Oats, rolled or flaked grains [1] | 1 kg | 2.416 | Basil [4] | 1 kg | 8280 |
| Olive oil, virgin [1] | 1 kg | 14431 | Margarine [4] | 1 kg | 4190 |
| Olives [2] | 1 kg | 4400 | Cackrey [4] | 1 kg | 576 |
| Onion [1] | 1 kg | 272 | Fish [5] | 1 kg | 1974 |
| Peanuts [2] | 1 kg | 3100 | Manioc (cassava) [2] | 1 kg | 564 |
| Pepperoni [4] | 1 kg | 4800 | Maize oil [1] | 1 kg | 2575 |
| Pineapple [1] | 1 kg | 255 | Ketchup [1] | 1 kg | 534 |
| Plum in syrup [4] | 1 kg | 2180 | Dry pasta [1] | 1 kg | 1849 |
| Plum [1] | 1 kg | 2180 | Apple or pear [2] | 1 kg | 700 |
| Pork [2] | 1 kg | 4800 | Egg [3] | 1 kg | 3300 |
| Potato flakes [4] | 1 kg | 1044 | Tomato sauce [2] | 1 L | 1069 |
| Potato [2] | 1 kg | 250 | Corn [2] | 1 kg | 900 |
| Rice [2] | 1 kg | 3400 | Mango [1] | 1 kg | 1600 |
| Rosemary [4] | 1 kg | 8280 | Lentil [1] | 1 kg | 5874 |
| Sesame [1] | 1 kg | 9371 | Raisin [1] | 1 kg | 2433 |
| Sugar [2] | 1 kg | 1500 | Condensed milk | 1 kg | 5000 |
| Sweet potato [1] | 1 kg | 383 | Soy sauce [2] | 1 L | 613 |
| Tea [2] | 1 cup of 250 mL | 30 | Pepper [1] | 1 kg | 7611 |
| Tomato extract [1] | 1 kg | 713 | Parsley [4] | 1 kg | 8280 |
| Wheat flour [1] | 1 kg | 1849 | Tomato [2] | 1 kg | 180 |
| Yam [1] | 1 kg | 343 | | | |
| Zucchini [4] | 1 kg | 353 | Orange [2] | 1 kg | 460 |

* GAF: global average footprint per litre; [1]—Mekonnen e Hoekstra (2011); [2]—Hoekstra (2008); [3]—Hoekstra (2010); [4]—author's adaptation; [5]—Pahlow et al. (2015).

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
