# Peer review of "Environmental Impacts of University Restaurant Menus: A Case Study in Brazil"

_sustainability, doi:10.3390/su11195157_

Round 1
Reviewer 1 Report
In the assessment of the paper submitted for the review, I specifically focused on the discussed issues, applied research procedure, substantive content of the paper and its structure.
The considerations conducted in the paper are focused on such categories as: foodservices, environmental impacts, water footprint, university restaurant menus in Brazil.
The subject area discussed in the paper is important and topical. It is also consistent with the profile of the Journal.
The value of the paper results from combination of literature studies with the results of Authors' own research.
However, deliberations conducted in the paper need to be expanded. Therefore, it is specifically recommended to:
- take into consideration more publications in the sphere of discussed subject matter,
- develop a literature review (as a separate part of article),
- develop the discussion of results and conclusions,
- specify the managerial implications,
- develop the description of the limitations of conducted research and indicate the trends for further empirical research.
Author Response
Response to Reviewer 1 Comments
Article:
“Environmental impacts of university restaurant menus: a case study in Brazil”.
Dear Reviewer,
We appreciate your suggestions for improving the manuscript. We would like to point out that we also made a change in water footprint table because the values sent earlier did not correspond to the per capita values for some foods. With the changes made, we believe that the article is now more clear and complete.
We list your suggestions below with our answers.
Point 1. In the assessment of the paper submitted for the review, I specifically focused on the discussed issues, applied research procedure, substantive content of the paper and its structure.
Point 2. The considerations conducted in the paper are focused on such categories as: foodservices, environmental impacts, water footprint, university restaurant menus in Brazil.
Point 3. The subject area discussed in the paper is important and topical. It is also consistent with the profile of the Journal.
Point 4. The value of the paper results from combination of literature studies with the results of Authors' own research.
However, deliberations conducted in the paper need to be expanded. Therefore, it is specifically recommended to:
Point 5 - take into consideration more publications in the sphere of discussed subject matter,
Response 5. We added new references on the subject covered in the article. We believe the manuscript has now a strong theoretical framework, as we cite important documents related to sustainable production in the food sector, as well as recent articles on the subject.Point 6- develop a literature review (as a separate part of article),
Response 6. We agree with your suggestion. We add a literature review in section 2 of the article.Point 7 - develop the discussion of results and conclusions,
Response 7. We agree. We made adjustments in the discussions of results and final considerations.
Point 8 - specify the managerial implications,
Response 8. Some managerial implications were added in the final considerations.
Point 9 - develop the description of the limitations of conducted research and indicate the trends for further empirical research.
Response 9. The limitations of the study were added in the final considerations
The changes made were highlighted in the source file in red.

Reviewer 2 Report
About the paper with the title "Environmental Impacts of University Restaurant Menus in Brazil: A Case Study" I have the following comments:
- This work have a clear language, however I consider this study very weak in the present version. In fact, to consider only 112 menus from only one restaurant and for only 2 months is very little for a scientific paper.
- However, considering that the sample is very weak you can invest in the literature review and in the statistical methodology for the data analysis.
- I suggest you improve significantly your paper with a good literature review and with good approach for the data analysis (regressions, factor analysis, cluster analysis, Classification trees, DEA, ......). If not your paper seems a technical report.
- The final considerations section (very poor and vague) is a consequence of a weak approach.
Author Response
Response to Reviewer 2 Comments
Article:
“Environmental impacts of university restaurant menus: a case study in Brazil”.
Dear Reviewer,
We appreciate your suggestions for improving the manuscript. We would like to point out that we also made a change in water footprint table because the values sent earlier did not correspond to the per capita values for some foods. With the changes made, we believe that the article is now more clear and complete.
We list your suggestions below with our answers.
Point 1. This work have a clear language, however I consider this study very weak in the present version. In fact, to consider only 112 menus from only one restaurant and for only 2 months is very little for a scientific paper.
Response 1: Thanks for the comment. However, we emphasize that the article is a case study. We emphasize that the restaurant serves approximately 4000 meals a day. The number of items that are required to perform the menu calculations should also be highlighted.
Point 2. However, considering that the sample is very weak you can invest in the literature review and in the statistical methodology for the data analysis.
Response 2: We have added new references on the subject covered in the article. We believe that now the manuscript has a strong theoretical framework, as we cite important documents related to sustainable production in the food sector, as well as recent articles on the subject. We add a literature review in section 2 of the article.Regarding statistical analysis, after confirming the normality of the data by the Shapiro-Wilk test we performed a T test to confirm the statistical difference between vegetarian and traditional menu.
Point 3. I suggest you improve significantly your paper with a good literature review and with good approach for the data analysis (regressions, factor analysis, cluster analysis, Classification trees, DEA, ......). If not your paper seems a technical report.
Response 3: We welcome your suggestion regarding literature review. Regarding data analysis we answered in point 2.
Point 4. The final considerations section (very poor and vague) is a consequence of a weak approach.
Response 4: We proceed with writing adjustments. As for your opinion "is a consequence of a weak approach": we respect, but we do not agree. There are no papers dedicated to making a Water Footprint (WF) menu evaluation considering the totality of foodstuffs in the international literature. We evaluated all food labels and invoices information to estimate the menu WF.
The changes made were highlighted in the source file in red.

Reviewer 3 Report
Comments:
1. This paper addressing an interesting topic with potential impact. It aims to evaluate Water Footprint (WF) of menus offered in a public university restaurant in the northeast of Brazil.
2. The title needs minor improvement and it should be revised to: “Environmental impacts of university restaurant menus: The case of Brazil”. The word “Enviromental” is spelled incorrectly as well.
3. The abstract should be improved by adding 1-2 warm-up sentences to introduce the main topic at the beginning.
4. The abstract should be enriched by highlighting the main conclusions of the study at the end as well.
5. Line 50 “Among the various types and types of catering services …”; the authors should add a paragraph and discuss various types of catering services briefly.
6. The authors should add a few relevant/recent studies along with their approaches and outcomes and indicate the main contribution of the current study by comparing it with previous ones in the Introduction section.
7. I miss more emphasis on the global novelty of the current study compared to similar studies; it should be done in the Introduction section in one paragraph.
8. Line 71 “…is to evaluate the water footprint (PH)…”; the abbreviation has been added mistakenly and should be revised to WF.
9. To enrich the main aims and objectives, I suggest adding a few research questions in the Introduction section and ensure that all these questions are properly addressed in the Conclusion section.
10. Overall, the Introduction section is rather weak and short with no concrete discussion on the main topic.
11. The title of sub-section “2.1. Study Caracterization” should be revised to “2.1. Study Characterization”.
12. To improve the Methods section, the authors should add the map of the study area to this section as well as adding short description regarding this area.
13. Line 97 “For data collection regarding the acquisition…” is not reading well; the authors should reformulate it to a fluent sentence.
14. Line 106 “For in nature foods that did not have labeled packages…” is very long and heavy. The authors should split it in two sentences.
15. Line 16 “For analysis of water footprint (WF)…”; when an abbreviation is defined in the beginning of the paper, there is no need to use the full term every time. The authors should carefully revise the whole paper.
16. Line 167 “This reality is built in the face of population…” is rather long. The authors should split it in two sentences.
17. Line 196, the authors should avoid adding very short paragraphs; it should be either enriched or merged with the next paragraph.
18. Line 201 “…the per capita value of PH…”; what is ‘PH’? It should be defined and explained.
19. There is no real Discussion section as the discussion of the results according and compared to existent literature is missing. The authors should outline how the main findings are in line with previous studies.
20. The authors failed in providing the Conclusion section as well. There is a need for such section to elaborate on the main (policy) implications of the findings.
21. The authors should highlight the future research directions in one paragraph in the Conclusion section.
22. The English grammar and style should be checked throughout the paper (especially long and heavy statements; typos and incorrect abbreviations).
Author Response
Response to Reviewer 3 Comments
Article:
“Environmental impacts of university restaurant menus: a case study in Brazil”.
Dear Reviewer,
We appreciate your suggestions for improving the manuscript. We would like to point out that we also made a change in water footprint table because the values sent earlier did not correspond to the per capita values for some foods. With the changes made, we believe that the article is now more clear and complete.
We list your suggestions below with our answers.
Point 1. This paper addressing an interesting topic with potential impact. It aims to evaluate Water Footprint (WF) of menus offered in a public university restaurant in the northeast of Brazil.
Response 1: Thank you. We believe that research evaluating environmental and social impacts in the foodservice industry should be encouraged. Usually only nutritional and food safety variables are taken into account.Point 2. The title needs minor improvement and it should be revised to: “Environmental impacts of university restaurant menus: The case of Brazil”. The word “Enviromental” is spelled incorrectly as well.
Response 2: We corrected the word. However, we find it more appropriate to maintain “a case study in Brazil”.Point 3. The abstract should be improved by adding 1-2 warm-up sentences to introduce the main topic at the beginning.
Response 3: According. We proceeded with the requested adjustment.
Point 4. The abstract should be enriched by highlighting the main conclusions of the study at the end as well.
Response 4: We proceeded with the requested adjustment.
Point 5. Line 50 “Among the various types and types of catering services …”; the authors should add a paragraph and discuss various types of catering services briefly.
Response 5: According. We have included the requested information in Literature Review section.
Point 6. The authors should add a few relevant/recent studies along with their approaches and outcomes and indicate the main contribution of the current study by comparing it with previous ones in the Introduction section.
Response 6: The requested approach was included in the Results and Discussion section.
Point 7. I miss more emphasis on the global novelty of the current study compared to similar studies; it should be done in the Introduction section in one paragraph.
Response 7: According. We proceeded with the requested adjustment.
Point 8. Line 71 “…is to evaluate the water footprint (PH)…”; the abbreviation has been added mistakenly and should be revised to WF.
Response 8: According. The requested correction was made.
Point 9. To enrich the main aims and objectives, I suggest adding a few research questions in the Introduction section and ensure that all these questions are properly addressed in the Conclusion section.
Response 9: According. We proceed the inclusion of 2 main topics at the final of introduction section. These 2 topics are addressed in conclusion.
Point 10. Overall, the Introduction section is rather weak and short with no concrete discussion on the main topic.
Response 10: According. We made adjustments in Introduction.
Point 11. The title of sub-section “2.1. Study Caracterization” should be revised to “2.1. Study Characterization”. According. The requested correction was made.
Response 11: According. The requested correction was made.
Point 12. To improve the Methods section, the authors should add the map of the study area to this section as well as adding short description regarding this area.
Response 12: We have included the requested information. We added a map of the region where the study was conducted and information about that region in the "Study Characterization" section. We also added a T test to compare the averages of menus Water Footprints.Point 13. Line 97 “For data collection regarding the acquisition…” is not reading well; the authors should reformulate it to a fluent sentence.
Response 13: According. We adjusted the wording of the sentence.
Point 14. Line 106 “For in nature foods that did not have labeled packages…” is very long and heavy. The authors should split it in two sentences.
Response 14: We adjust as requested.
Point 15. Line 16 “For analysis of water footprint (WF)…”; when an abbreviation is defined in the beginning of the paper, there is no need to use the full term every time. The authors should carefully revise the whole paper.
Response 15: We agree with the observation and review the text.
Point 16. Line 167 “This reality is built in the face of population…” is rather long. The authors should split it in two sentences.
Response 16: Request accepted.
Point 17. Line 196, the authors should avoid adding very short paragraphs; it should be either enriched or merged with the next paragraph.
Response 17: According. We adjust as requested.
Point 18. Line 201 “…the per capita value of PH…”; what is ‘PH’? It should be defined and explained.
Response 18: We proceed with the correction for "WF".
Point 19. There is no real Discussion section as the discussion of the results according and compared to existent literature is missing. The authors should outline how the main findings are in line with previous studies.
Response 19: According. We proceeded with the requested revision.
Point 20. The authors failed in providing the Conclusion section as well. There is a need for such section to elaborate on the main (policy) implications of the findings.
Response 20: We proceed with writing adjustments.
Point 21. The authors should highlight the future research directions in one paragraph in the Conclusion section.
Response 21: According. We adjust as requested.
Point 22. The English grammar and style should be checked throughout the paper (especially long and heavy statements; typos and incorrect abbreviations).
Response 22: We inform you that we proceed with the revision of the text and grammar with a British English native.
The changes made were highlighted in the source file in red.

Round 2
Reviewer 1 Report
Thank You for introducing my suggestions and comments in article.
Author Response
Dear Reviewer,
We are very grateful for your contribution to improving the manuscript.
Reviewer 2 Report
The literature review needs to be significantly improved and the final considerations section should be rewritten with the new insights brought by this research.
Author Response
Dear Reviewer,
We appreciate your suggestions for improving the manuscript.
We have added some information in the Literature Review section. We arranged the final considerations to make the information clearer and added a comment. We put the changes made in round 2 in blue.
Reviewer 3 Report
No further comments.
Author Response

(The authors gave the same response as above.)
